# AdaptiveStep: Automatically Dividing Reasoning Step through Model Confidence

**Yuliang Liu** [* 1 2] **Junjie Lu** [* 3] **Chaofeng Qu** [4] **Zhaoling Chen** [1] **Zefan Cai** [5] **Jason Klein Liu** [4] **Chonghan Liu** [4] **Yunhui Xia** [4] **Li Zhao** [6] **Jiang Bian** [6] **Chuheng Zhang** [6] **Wei Shen** [4] **Zhouhan Lin** [2 7]

## Abstract

Current approaches for training Process Reward Models (PRMs) often involve decomposing responses into multiple reasoning steps using rule-based techniques, such as using predefined placeholder tokens or setting the reasoning step's length to a fixed size. These approaches overlook the fact that certain words don't usually indicate true decision points. To address this, we propose AdaptiveStep, a method that divides reasoning steps based on the model's confidence in predicting the next word, offering more information on decision-making at each step, improving downstream tasks like reward model training. Moreover, our method requires no manual annotation. Experiments with AdaptiveStep-trained PRMs in mathematical reasoning and code generation show that the outcome PRM achieves state-of-the-art Best-of-N performance, surpassing greedy search strategy with token-level value-guided decoding, while also reducing construction costs by over 30% compared to existing open-source PRMs. We also provide a thorough analysis and case study on its performance, transferability, and generalization capabilities. We provide our code on https://github.com/Lux0926/ASPRM.

## 1. Introduction

Large language models (LLMs) have demonstrated exceptional performance across various tasks. However, even most advanced LLMs struggle to generate correct solutions when facing complex reasoning problems, such as mathe-

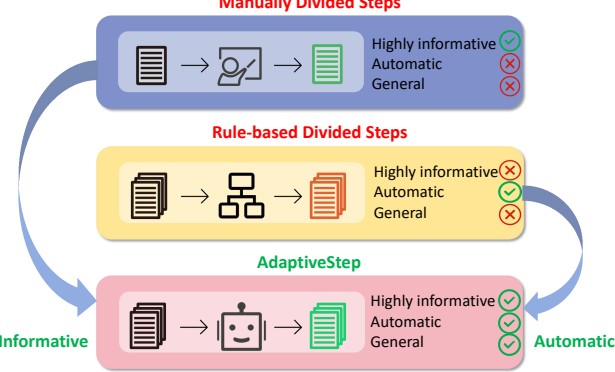

Figure 1: Rule-based reasoning step dividing (e.g., using line breaks or a fixed number of tokens) is automated but results in low informativeness at the end of the step and is difficult to apply in domains that are hard to define rules. In contrast, manual step division provides high informativeness but is costly to scale and heavily reliant on the experts' domain knowledge. AdaptiveStep, which divides steps based on model confidence, addresses these challenges by offering automation, efficiency, high informativeness, and applicability across various domains.

matical reasoning and code generation tasks (Huang et al., 2024; Tyen et al., 2024; Mirzadeh et al., 2024; Shen & Zhang, 2024). To address these challenges using the stepwise Chain of Thought (CoT) approach (Wei et al., 2023), various strategies have been proposed by the research community (Qin et al., 2024; DeepSeek-AI et al., 2025; Team et al., 2025). One promising method is training Process Reward Models (PRMs), which offer more fine-grained rewards at each reasoning step compared to Outcome Reward Models (ORMs), guiding the LLM to generate higher-quality responses than the original model output (Shao et al., 2024; Sessa et al., 2024; Gao et al., 2024).

However, as illustrated in Figure 1, existing PRMs typically divide a model's response into multiple reasoning steps using rule-based methods, such as chopping with a pre-defined symbol. This results in a series of coarse reasoning step divisions that lack decision-making information at steps (Wang et al., 2024a; Lightman et al., 2023). Moreover, rule-based methods also face challenges when applied to tasks where

---

[*]Equal contribution [1]Nanjing University [2]Shanghai Innovation Institute [3]University of Technology Sydney [4]Independent [5]UW-Madison [6]MSRA [7]Shanghai Jiaotong University. Correspondence to: Chuheng Zhang <zhangchuheng123@live.com>, Wei Shen <shenwei0917@126.com>, Zhouhan Lin <lin.zhouhan@gmail.com>.

*Proceedings of the 42nd International Conference on Machine Learning*, Vancouver, Canada. PMLR 267, 2025. Copyright 2025 by the author(s).

the steps are difficult to define. Some studies have explored the application of PRMs at the level of individual tokens or a fixed number of tokens (Lee et al., 2024; Luo et al., 2024); nevertheless, balancing annotation costs with the granularity of division remains a challenge. Although studies have demonstrated the advantages of PRMs over ORMs, these limitations, along with the high building costs, continue to constrain the broader adoption of PRMs (DeepSeek-AI et al., 2025).

To address these issues, we aim to find an automatic step-dividing method to divide reasoning solutions into more informative steps, in contrast to the coarse division by rule-based methods. As suggested by Kahneman (2011), the cognitive cost of reasoning varies depending on the difficulty of the decision or task. Additionally, a statistical analysis of common errors in reasoning tasks conducted by Roy & Roth (2016) revealed that many errors stem from incorrect numerical calculations or the misapplication of words, particularly verb misuse. This suggests that certain types of words or positions in the reasoning process require more attention.

Therefore, our goal is to divide the reasoning responses at these key positions to ensure the valuable costs during inference and training. We find that by pivoting on the prediction confidence, the model can automatically identify the critical breaking points in the reasoning process. Accordingly, we propose AdaptiveStep, a method that divides reasoning steps based on model confidence (Hills & Anadkat, 2024). We conduct experiments on the PRM scenario, with the resulting PRM named the AdaptiveStep Process Reward Model (ASPRM). This dividing method yields highly informative step divisions, enabling downstream tasks (e.g., processing the reward model) to enhance performance.

In our experiments, we assess the effectiveness of ASPRM in mathematical reasoning and code generation tasks using the Best of N (BoN) evaluation. For the mathematical reasoning task, we evaluate on GSM8k (Cobbe et al., 2021) and MATH500 (Lightman et al., 2023) dataset. For the code generation task, we collect a dataset named **LeetCodeDataset** containing 1,940 problems from LeetCode, along with the corresponding Python solutions and test cases, which include training and test splits to train and evaluate the PRM and further assess it on the LiveCodeBench (Jain et al., 2024).

Additionally, the most widely used PRM step-dividing method relies on fixed symbols, limiting the accuracy of the more fine-grained judgment ability of PRMs. We find that ASPRM can provide precise rewards to perform Token-level Value-guided Decoding (TVD) for reasoning tasks, offering another evaluation method by integrating PRM directly into the model inference process.

In mathematical reasoning tasks, ASPRM outperforms pre-

vious open-source methods in BoN evaluation. In addition, compared to greedy decoding, TVD further improves the final performance by 3.15% and 14.4% on the GSM8k and MATH500 datasets, respectively, while incurring less than 70% of the training data construction costs compared to the open-source baselines. In code generation tasks, ASPRM shows superior performance and robustness in BoN evaluation compared to ORM. It outperforms greedy decoding by 6.54% and 3.70% on the two datasets in TVD evaluation.

Our main contributions are as follows:

1. We propose an automatic, efficient, general, and highly informative reasoning step-dividing method, AdaptiveStep, along with its corresponding PRM implementation.

2. Our results show that ASPRM is currently the state-of-the-art PRM, empirically simple and low-cost training data construction. Furthermore, the PRM built using AdaptiveStep demonstrates stronger discriminative power at the token level compared to greedy search and existing methods. Additionally, we analyze and explore several properties of ASPRM, including transferability, domain generalization, and division features of the training data.

3. We open-source a collection of competition-level coding problems from LeetCode, along with test cases, and provide an easy-to-use sandbox. We also release the dataset, models, and our code.

## 2. Related Works

**Step-wise methods for LLMs reasoning:** Chain-of-Thought (CoT) (Wei et al., 2023) reasoning has become a foundational approach in LLM reasoning. Scaling the number of tokens and steps in test time to tackle complex problems has become common practice (Team et al., 2025; DeepSeek-AI et al., 2025). In this paradigm, the model generates an intermediate step-wise solution before providing a final answer. As expectations for model performance on more complex tasks increase, methods for step-wise verification and alignment have also advanced rapidly, like PRM (Zhang et al., 2024; Wang et al., 2024a; Yuan et al., 2024) and step-wise RLHF (Chen et al., 2024; Lai et al., 2024; Wang et al., 2024b). Inference time step-wise methods also significantly enhance the model's reasoning capabilities, such as Monte Carlo methods (Feng et al., 2023), step-wise self-consistent (Zhao et al., 2024), step-wise beam search (Lee et al., 2024), and flexible divide-and-conquer methods (Yao et al., 2023; Hao et al., 2023) for planning.

**PRM for LLM reasoning and step-dividing methods:** The importance of intermediate reasoning steps in LLMs

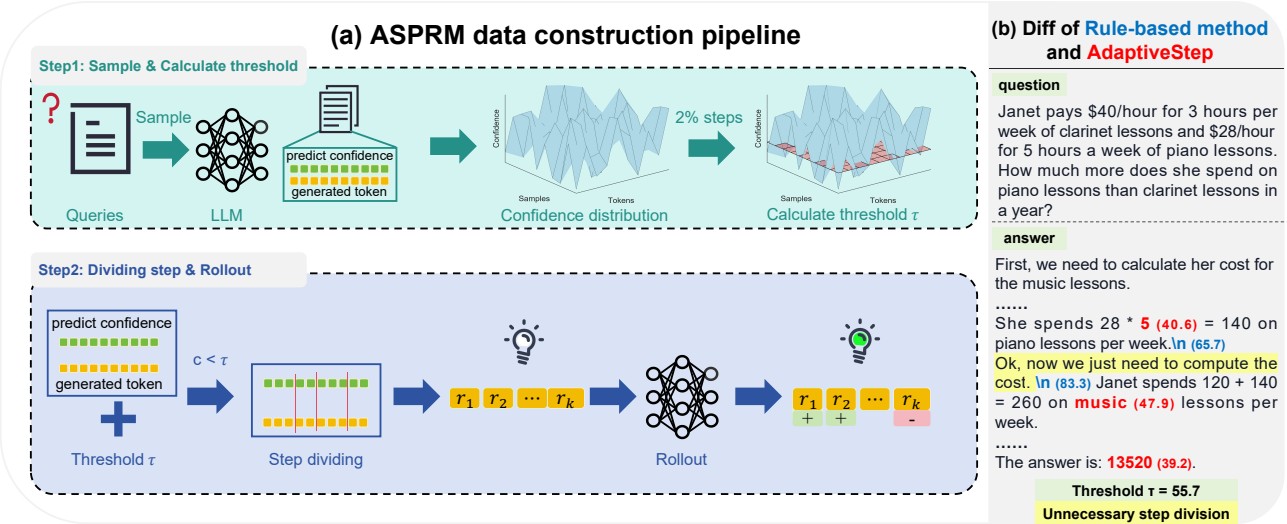

Figure 2: Method overview. **a) ASPRM Training Data Construction Pipeline.** Step 1: Sample from the dataset of a given domain, collecting confidence scores and samples for the training data. Then, accumulate the confidence distribution of all samples and determine the threshold. Step 2: Divide reasoning steps based on the threshold and label the steps using rollout. **b) The difference between Rule-based method and AdaptiveStep division.** The Rule-based method divides the reasoning process using predefined symbols or fixed token counts (e.g., line breaks, as shown in the figure), while AdaptiveStep divides reasoning steps based on model confidence. We observe that the model tends to divide reasoning steps at key decision points, such as within mathematical expressions, at noun selections, and when determining the final answer. In contrast, we find that the confidence at line breaks is particularly high.

for complex tasks was highlighted by Uesato et al. (2022), which led to the development of Process Reward Models (PRMs) to enhance LLM reasoning by providing feedback at each step. Lightman et al. (2023) showed that step-by-step feedback improves reasoning reliability and reduces logical errors. Similarly, the OmegaPRM (Luo et al., 2024), utilizing Monte Carlo Tree Search (MCTS), improves mathematical reasoning performance by efficiently gathering process supervision data. Wang et al. (2024a) proposed a heuristic annotation method, reducing PRM annotation costs. Step-level reward models (Ma et al., 2023) have demonstrated that feedback at each step helps guide LLMs to more optimal solutions. Automated process verifiers (Setlur et al., 2024) further enable large-scale deployment of PRMs, improving LLM alignment. Several works have explored PRM applications in reasoning tasks (Xia et al., 2024; Ma et al., 2023; Luo et al., 2023; Snell et al., 2024). However, the predominant step-dividing method used in PRMs or other step-wise methods remains rule-based, such as using pre-defined symbols, which results in sentence-level PRMs. Some works have developed token-level PRMs by dividing at fixed token intervals, but the high annotation cost remains a limitation (Lee et al., 2024; Luo et al., 2024).

**Guided decoding:** Standard decoding in Large Language Models (LLMs) typically involves sampling strategies to select the next token. Guided decoding has been widely ex-

plored to improve performance and constrain text generation. Chaffin et al. (2021) proposed incorporating a value model into the LLM decoding process, using MCTS (Coulom, 2006) to constrain output without fine-tuning. Liu et al. (2024) integrated the Proximal Policy Optimization (PPO)-based value network with MCTS, enabling collaboration with the policy network during inference. In the code generation domain, Planning-Guided Transformer Decoding (PG-TD)(Zhang et al., 2023) uses planning algorithms for lookahead search to guide the transformer in producing more optimal code. Nie et al. (2024) employed a proxy code LLM to build an offline token-scoring model that reallocates token probabilities to guide decoding. Additionally, several works have applied value functions to guide token-level decoding (Dathathri et al., 2019; Choi et al., 2023; Xu et al., 2024; Krause et al., 2020). In this paper, we use PRM as a value function to directly guide the decoding process of large language models, aiming to validate the effectiveness of PRM and explore additional potential applications of PRM.

## 3. Methods

In this section, we first introduce how AdaptiveStep divides responses into reasoning steps, and then present how a PRM can be trained on these data, as shown in Figure 2. At last, we introduce Token-level Value-guided Decoding (TVD)

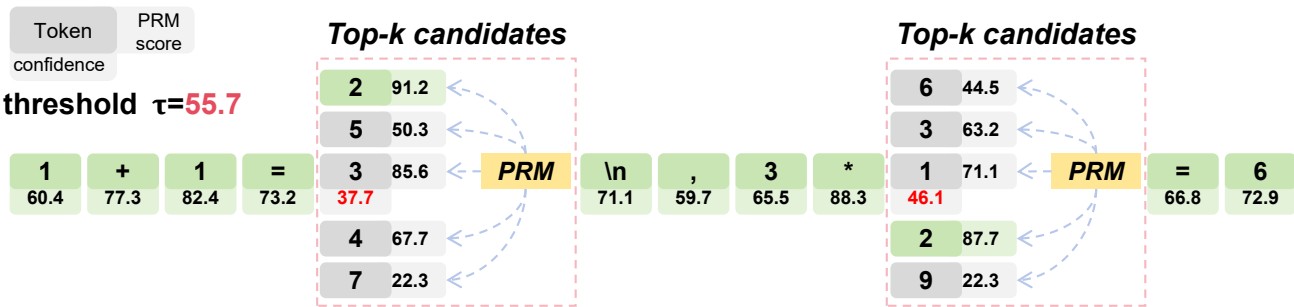

Figure 3: We illustrate Token-level Value-guided Decoding (TVD) with a simple example. The green token denotes the selected tokens, while the gray token indicates the tokens that were not selected. The question is *3 * (1 + 1) = ?*, and the correct output is *6*. In this case, the model exhibits low confidence (where $c_y < \tau$) when calculating the result of 1+1, and subsequently determines which number to multiply by 3. The PRM should select the best token based on its judgment to arrive at the correct final answer. As shown in the top-left corner, for each token, the middle box represents the token itself, the bottom box shows the predicted confidence, and the box on the right displays the PRM score. The red confidence score indicates that the confidence of the Top-1 predicted candidate is lower than the threshold.

that can get better responses using the trained PRM.

### 3.1. AdaptiveStep

Given a question $q \in Q$, we can generate $N$ responses with temperature-based random sampling using the language model $\pi$. We denote generated responses as $\{s^1, s^2, \cdots, s^N\}$ with $s^n \in S$. (For ease of notation, we omit the dependence of the response $s^n$ on $q$.) In this way, we obtain a set of question-response pairs $(Q \times S)$.

To divide the responses into reasoning steps, we use the probability of the sampled token as the metric for *model confidence* (Hills & Anadkat, 2024). Then we determine a threshold $\tau$, which is based on a certain proportion of the token count, such that the tokens below this threshold become a breaking point.

Specifically, the model confidence can be written as

$$c_{s_i^n} = p(s_i^n | \pi, q, s_{<i}^n) \tag{1}$$

where we use $s_i^n$ and $s_{<i}^n$ to denote the $i$-th token and the tokens prior to the $i$-th token in the response, respectively. Low model confidence at the $i$-th token indicates that the model is hard to determine the token selection at the $i$-th position, and therefore, this position may become the starting point of a new reasoning step.

According to the above procedure, we divide the response $s^n$ into $K$ reasoning steps $s^n = \{r_1, r_2, ..., r_K\}$ where the last token within each reasoning step is associated with below-the-threshold model confidence.

### 3.2. PRM Training

To train a PRM based on the question-response pairs with divided reasoning steps, we first need to estimate the target

reward for each reasoning step and then train a PRM that can predict the reward.

To estimate the target reward, we mainly follow the heuristic rollout method proposed by Wang et al. (2024a). We rollout the response generation process $J$ times starting from each reasoning step, resulting in rollouts denoted as $\{p, r_1, ..., r_k, t_j\}_{k \in [K], j \in [J]}$, where $t_j$ is the $j$-th trajectory starting from a partial response.

Then, we estimate the target reward of this step based on the correctness of any decoded solution. We use hard estimation (HE) to estimate the reward for the step $r_k$. HE indicates whether any of the responses starting from the current partial response can reach a correct answer. For our implementation, in the code generation tasks, we define correctness as whether the solution can pass all test cases; in the math reasoning tasks, we define correctness as whether the answer matches the ground truth. Formally, the target reward can be estimated as

$$r_k^e = \begin{cases} 1, & \exists j \in [J], \{r_1, ..., r_k, t_j\} \text{ is correct} \\ 0, & \text{otherwise} \end{cases} \tag{2}$$

With the target rewards estimated based on the rollouts, we can train PRM using the following loss:

$$\mathcal{L}_{PRM}^\theta = -\sum_{k=1}^{K} (r_k^e \log r_k^\theta + (1 - r_k^e) \log(1 - r_k^\theta)), \tag{3}$$

where $r_k^e$ is the target reward and $r_k^\theta := R^\theta(p, r_1, \cdots, r_k)$ denotes the reward predicted by the PRM $R^\theta$.

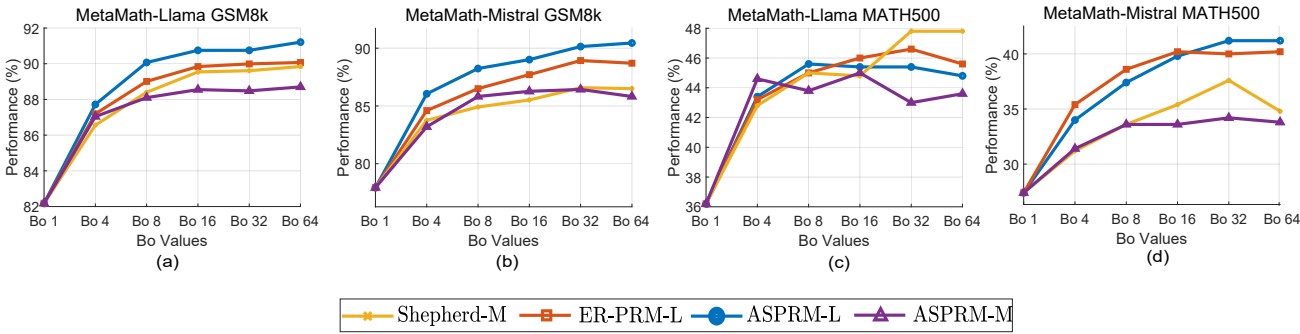

Figure 4: BoN results for the math tasks. We evaluate all PRMs on: (a) MetaMath-Llama generated GSM8k candidate solutions; (b) MetaMath-Mistral generated GSM8k candidates; (c) MetaMath-Llama generated MATH500 candidates; and (d) MetaMath-Mistral generated MATH500 candidates. The "-L" and "-M" suffixes denote the base models (Llama and Mistral, respectively). We report the evaluation results based on the released versions of other works.

### 3.3. Token-level Value-guided Decoding

The TVD strategy leverages the PRM to guide token selection during language model decoding. Specifically, when the model encounters a low confidence score (Top-1 probability $c_p < \tau$) in decoding, it triggers the PRM to evaluate the tokens associated with the highest $M$ probability given by the policy model $\pi$: $\mathbf{s_i^*} = \{s_i^1, s_i^2, \ldots, s_i^M\}$.

Among these candidates, the PRM selects the token it considers the best based on its learned reward estimation mechanism:

$$s_i = \arg\max_{s_i^m \in s_i^*} R^\theta\left(p, s_{<i}, s_i^m\right), \qquad (4)$$

Where $s_i$ is the token selected as the optimal choice for the low-confidence decoding position, and $R^\theta(\cdot)$ represents the score given by the PRM.

## 4. Experiments and Analysis

In this section, we first show our experiment setup, including the dataset usage, model selection, baselines, metrics, and parameter setup. We then present the experimental results, followed by an analysis of the transferability, generalization, and features of the division.

### 4.1. Experiments Setup

**Datasets and models:** We use MetaMathQA (Yu et al., 2023) to train Mistral-V0.1 (Jiang et al., 2023) (Mistral), which is termed MetaMath-Mistral, to serve as $\pi$ in math domain, and use LeetCodeDataset[1] training data to train

Deepseek-Coder-Base (Guo et al., 2024), which is called LCD-DS to serve as $\pi$ in the code domain. To get the math PRM training data, we sample the MATH and GSM8k training datasets using MetaMath-Mistral and sample LeetCode-Dataset training data using LCD-DS to generate code PRM training data. For evaluation, we use MATH500, the GSM8k test set, the LeetCodeDataset test set, and LiveCodeBench-V4 (Jain et al., 2024). To align with previous work and conduct further analysis, we train two math PRMs: ASPRM-L (based on Meta-Llama-3.1-8B (Grattafiori et al., 2024), which is called Llama in the following) and ASPRM-M (based on Mistral-V0.1), both with MetaMath-Mistral generated training data. And one code PRM: ASPRM-D (based on DeepSeek-Coder-Base) with LCD-DS generated data.

**Baselines and metrics:** There are several open-sourced PRMs in the math domain. We select Math-Shepherd (Wang et al., 2024a) and ER-PRM (Zhang et al., 2024) as our baselines. For the code domain, due to the limited availability of open-source code PRMs with competitive construction costs, we trained a code ORM as a baseline using the same data, with only the final rating position considered.

For all tasks, we evaluate the PRMs' performance using the Best of N (BoN) metric and further assess model capabilities with TVD. In math reasoning tasks, we evaluate whether the model's final answer matches the ground truth exactly. In the code tasks, we test the generated code by running it in a sandbox and checking if it passes all test cases. Following Wang et al. (2024a), we use the minimum PRM score across all scored steps as the PRM's final judgment for a given candidate in BoN.

**Parameter Settings:** We sample 30 times per data point and deduplicate the responses in Step 1. For labeling the

---

[1]To train ASPRM for code tasks, we collected 1,745 problems from the *LeetCode problems* as our training set and 175 problems as the test set. The test cases for these data are manually collected from the LeetCode website (excluding the test cases within the problem). The solutions are gathered from GitHub open-sourced repositories, mainly from

https://github.com/doocs/leetcode, checked by GPT-4, and cross-verified with the test cases.

Table 1: Token-level Value-guided Decoding results. A/P@1 refers to the inference model's greedy search performance, we use Accuracy@1 for math tasks and Pass@1 for code tasks as the metrics. ↑ and ↓ represent the performance improvement or decline compared to A/P@1.

| Dataset | Inference Model | A/P@1 | Math-Shepherd | ER-PRM | ASPRM-L / -M | ASPRM-D |
|---------|----------------|-------|---------------|--------|--------------|---------|
| **GSM8k** | MetaMath-M | 77.10 | 75.66↓ | 75.13↓ | **79.53**↑ / 77.33↑ | / |
| | MetaMath-L | 81.80 | 81.73↓ | 81.58↓ | **83.47**↑ / 82.56↑ | / |
| **MATH500** | MetaMath-M | 25.00 | 27.60↑ | 27.80↑ | **28.60**↑ / 26.80↑ | / |
| | MetaMath-L | 38.80 | 41.00↑ | 38.60↓ | **42.00**↑ / 41.20↑ | / |
| **LeetCodeDataset** | LCD-DS | 26.28 | / | / | / | 28.00↑ |
| **LiveCodeBench** | LCD-DS | 19.21 | / | / | / | 19.92↑ |

PRM training data, we perform 8 rollouts per step using the same model $\pi$. This process generates 388k PRM training samples. We use MetaMath-Mistral-generated data to train the math PRM. And we get 49k PRM samples for the code PRM. In our PRM training data, each sample includes a labeling point at the end of the response. We divide the responses by 2% The value is set according to Kahneman (2011), which finds that deep thinking for humans accounts for 2% of the total thinking.

## 4.2. Overall Results

**BoN Results**   We report the BoN evaluation results for the math dataset in Figure 4, and for the code dataset in Figure 5, respectively.

In the math tasks, ASPRM-L performs best across Figure 4(a), 4(b), and 4(d) despite under more stringent conditions: the training data sources, and the construction costs and models. 1) For **the training data sources**, AS-PRM only utilizes the GSM8k and MATH training sets during training data construction, while both ER-PRM and Math-Shepherd used the MATH test set (without using MATH500), which results in our performance being inferior to theirs on MATH500. 2) For **the costs and models used in construction**, the data construction costs for AS-PRM is less than 70% of that for the other two methods and only used a single construct model. In addition to the above problems that lead ASPRM-M to poor performance in the MATH500 dataset, we attribute its performance in Figure 4(a) to the training dataset being constructed by a single model, constraining its test-time transferability.

In the code tasks results shown in Figure 5, ASPRM-D demonstrates superior judgment ability. As N increases, the robustness of ASPRM-D outperforms that of ORM.

**TVD Results**   We report TVD results in Table 1. In the math reasoning task, ASPRM has consistently shown an ability to enhance the reasoning capacity of the inference models. While the performance guided by ER-PRM and

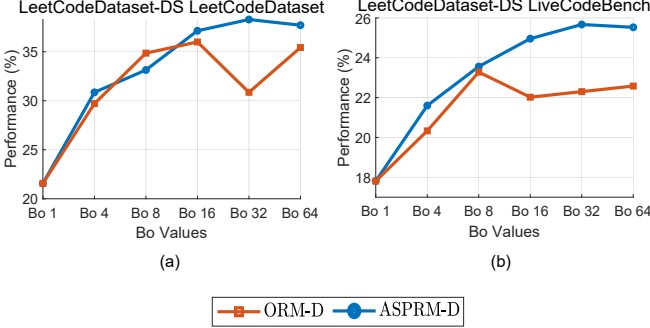

Figure 5: BoN results for the code datasets, we test ASPRM-D and a Code-ORM (ORM-D) on (a) LCD-DS generated LeetCodeDataset BoN candidates; (b) LCD-DS generated LiveCodeBench BoN candidates.

Math-Shepherd does not always demonstrate improvement, we hypothesize this is due to that the inference models already perform well with the greedy search on GSM8k, needing a more precise score to provide better guidance. The results further demonstrate the accuracy of the token-level judgment of ASPRM. In the code generation task, ASPRM has also achieved results surpassing greedy search by providing accurate judgment.

## 4.3. Transferability and Generalization Analysis

In this part, we investigate whether ASPRM demonstrates model transferability and rating position, in-domain, and cross-domain generalization capability, and the performance of mixed-domain data-trained PRM. In our experiments, unless otherwise specified, the BoN candidate generator and TVD inference model is MetaMath-Mistral.

**ASPRM exhibit model transferability:**   Since the quality of training data generated by rollout depends on the policy $\pi$, we explore the transferability of training data of our method. We get 371k PRM training data generated by MetaMath-Llama and conduct the same process as MetaMath-Mistral.

In Table 2, we find that training Mistral-V0.1 on data generated by MetaMath-Llama retains judgment ability, but its performance is weaker than that trained on data generated by the weak MetaMath-Mistral. This suggests that data generated through rollout has reasonable but limited transferability. Using multiple models for data construction, as in the Math-Shepherd, may be an effective strategy to enhance transferability.

Table 2: Transferability of PRM training data: **L to M** indicates training Mistral using PRM training data generated by MetaMath-Llama. ↑ and ↓ denote performance improvement or decline compared to ASPRM-M.

| Setup | Test Dataset | Bo64 / TVD |
|-------|-------------|------------|
| | M-MATH500 | 34.20↓ / 27.60↑ |
| L to M | M-GSM8k | 83.40↓ / 77.94↑ |
| | L-MATH500 | 41.80↓ / 41.40↑ |
| | L-GSM8k | 87.87↓ / 82.49↓ |

**ASPRM exhibit rating position generalization:** We evaluate the rating position generalization of different PRMs and show the results in Table 3. Three setups are employed in our experiments: **confidence**, **random**, and **hard**, we explain the setting in the caption of Table 3. The performance of ER-PRM-L shows a significant difference between the hard and random setups, whereas the difference of ASPRM-L between the two setups is relatively small. Additionally, ASPRM-M performs better under the *random* setup than under the *confidence* setup, demonstrating its superior generalization ability in the rating position. We attribute this advantage to the diversity of rating point types in the ASPRM training data.

Table 3: Rating position generalization. In the **confidence** setup, rating points are the positions where confidence falls below the threshold. In the **random** setup, rating points are selected at five random positions. In the **hard** setup, rating points are line breaks.

| Models | Scoring Setup | Bo64 |
|--------|--------------|------|
| ASPRM-L | confidence | 90.45 |
| | random | 90.22 |
| ASPRM-M | confidence | 85.82 |
| | random | 86.96 |
| MS-M | hard | 86.50 |
| | random | 86.20 |
| ER-PRM-L | hard | 88.70 |
| | random | 87.71 |

**ASPRM exhibit in-domain generalization:** We use **GSM-Symbolic** (Mirzadeh et al., 2024), which modifies variables or sentences in the original GSM8k dataset, to test whether PRM can achieve in-domain generalization. We show our results in Table 4. We find that ASPRM exhibits strong in-domain generalization as it achieves better results in TVD than greedy search, and selects the right samples in Bo64.

Table 4: In-domain generalization ability. The experiments are conducted on the GSM-Symbolic p2 dataset. ↑ indicates the performance improvement compared to greedy search.

| PRM Model | Base | Bo64 / TVD |
|-----------|------|------------|
| ASPRM-L | 22.80 | 51.56 / 24.56↑ |
| ASPRM-M | 22.80 | 37.88 / 24.68↑ |

**ASPRM exhibit cross-domain generalization:** We assess the cross-domain generalizability of PRMs using two setups: evaluating the math PRM in the code datasets and evaluating the code PRM in the math datasets. Our results are shown in Figure 5. We find that the ASPRM-L provides applicable guidance on code tasks and makes correct selections in BoN. However, ASPRM-D performs better on the more difficult MATH500 task but struggles on simple GSM8k. We hypothesize this is due to the long training data and long prompt in code PRM, as the GSM8k test data has a total length similar to the length of the prompt part of the code data on average, resulting in fewer low-confidence points for the model to learn.

Table 5: Cross-domain generalization ability of the PRMs: **Source** represents the source domain and the corresponding model. **Target** represents the target dataset domain and the corresponding test data. ↑ and ↓ indicate performance improvements or declines compared to the A/P@1 performance in Table 1.

| PRM Model | Target | Bo64 / TVD |
|-----------|--------|------------|
| ASPRM-L | Code-LCD | 34.29↑ / 28.00↑ |
| | Code-LCB | 22.30↑ / 19.21- |
| ASPRM-D | Math-GSM8k | 75.13↓ / 75.28↓ |
| | Math-MATH500 | 30.00↑ / 26.00↑ |

**Mixed data benefits downstream performance:** Since both tasks are reasoning tasks, we explore whether mixing training data from different domains can enhance downstream performance. To this end, we conduct two experiments: 1) training Mistral on a mixed math and code dataset, and evaluating it on MATH500 and GSM8k; 2) training DeepSeek on an equal amount of randomly sampled math and code data, and evaluating it on LeetCodeDataset and LiveCodeBench. The results are shown in Table 6. We find that mixing data improves PRM performance on math

datasets, while on code datasets, performance improves only in the TVD scenario on LiveCodeBench. We hypothesize this outcome is due to the following reason: for the math PRM, mixing long code domain training data improves the PRM's judging ability. For the code PRM, code domain training data is more difficult to obtain. Adding new data doubles the dataset size but introduces shorter data. This results in decreasing the global rating ability relied upon by BoN while enhancing the local rating ability used by TVD.

Table 6: The test results of the PRMs trained with a mixed training dataset. When the base model is Mistral, the *M+C* training data consists of the MetaMATH-Mistral generated math dataset and full code training dataset. When the base model is Deepseek, the *C+M* training data includes all of the code dataset and an equal amount of randomly sampled math training data. ↑ and ↓ represent the performance improvement or decline compared to the no mixed data trained PRMs in the origin domain of test data.

| Base Model | Train | Test | Bo64 / TVD |
|---|---|---|---|
| Mistral | M+C | GSM8k | 86.35↑ / 77.79↑ |
| | M+C | MATH500 | 35.40↑ / 29.00↑ |
| Deepseek | C+M | LCD | 37.71- / 28.00- |
| | C+M | LCB | 24.96↓ / 20.33↑ |

### 4.4. Threshold Analysis

In this part, we show the impact of different thresholds in dividing steps of our PRM training data. We add the BoN results of ASPRM models trained with the threshold of 0.5%, 1%, and 1.5% and test them on the GSM8k dataset with more task models, we show the results in Figure 6. However, larger thresholds (more than 3%) mean more rollouts than the baselines, so we only do the experiments with a threshold under 2%. We use multi-scale generators to test these PRMs.

We find that the PRM judgment ability is enhanced with the increase of dividing points in Figure 6 (a), but it is not significantly improved in Figure 6 (b). This shows that for models with different abilities, the optimal threshold choice may not be 2%, and more powerful models need less training data. Additionally, we find that at nearly 50% of the data points, ASPRM that using a single model to generate training data performs better than the baselines that use multiple training data generators.

### 4.5. Feature Analysis

In this part, we discuss the features of the AdaptiveStep division used in ASPRM and its advantages.

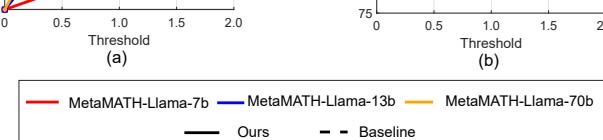

Figure 6: Threshold analysis of BoN results on GSM8k dataset with multi-scale BoN candidates generators. (a) Ours represents using Mistral as the PRM base model, and baseline represents Math-Shepherd; (b) Ours represents using Llama as the PRM base model, and baseline represents ER-PRM.

**Construction efficiency:** The training data construction of ASPRM demonstrates superior efficiency in both domains. In the math domain, the training data for ASPRM is generated using only a single MetaMath-Mistral model, with 30 samples per data point and 8 times rollouts per step. In contrast, ER-PRM performs fewer samples but conducts 16 times rollouts, while Math-Shepherd uses multiple models for sampling and rollouts. The average number of steps per sample and sample counts for each method are presented in Appendix A.2. As a result, the data construction costs for ASPRM are less than 70% of those for the other two. In the code domain, there are 14.4 lines on average per answer for the LeetCodeDataset training set, whereas only 5.69 steps are required for our method on average.

**Statistical features of the division:** There are several features and findings in the AdaptiveStep division statistics. For brevity, we refer to low-confidence tokens as "decision tokens" throughout this section. Taking the math PRM training data generated by Mistral as an example: 1) 3.85% tokens in mathematical expressions contribute 21.03% decision tokens; 2) only 2.7% decision tokens are newline tokens; 3) the inference model exhibits low confidence at semantic word points, particularly at Conjunction (29.00%), suggesting that continuous or transitional thinking is particularly challenging for the model.

For the code PRM training data: 1) the majority of decision points occur in the Code Comment type (80%), compared to the Code type (20%), even though Code Comments tokens account for only 19% of the total tokens; 2) a detailed analysis reveals that the Code Comment samples primarily fall into two subtypes: explaining what previous lines do and planning what to do in the following lines. The first subtype accounts for 9% of the samples, while the second accounts

for 91%. This indicates that the inference model triggers more during the planning process than during the writing process when generating code; 3) by further analyzing the Code type, we find that *Logical Operators*, *Block Begin Keyword*, *Control Statements* and *Loop Statements* occupy a high proportion of low confidence proportion with a small number of tokens. This suggests that, in addition to pre-planning in Comment, the model still requires assistance at certain logical decision points during the writing process.

The statistical information indicates that the inference model is prone to performing low confidence in the calculation process, semantic word selection in mathematics reasoning tasks, and the planning process in code generation tasks. The full statistical results are provided in Appendix A.1.

Our results in 4.3 indicate that PRM trained on mixed datasets can enhance downstream performance, making it possible to achieve better results in domains with hard-to-obtain data, such as code generation, at a lower cost. Based on the results and feature analysis in the code data, we hypothesize that the mutual enhancement arises from both tasks being reasoning problems. Similar to the text reasoning process in mathematics, the Code Comments contain substantial content that outlines subsequent steps. Therefore, training on a mixture of both datasets allows the model to achieve improved results.

## 5. Conclusion

In this paper, we propose a new reasoning step dividing method, AdaptiveStep, along with a corresponding Process Reward Model (PRM), ASPRM. We test the effectiveness of the PRM on mathematical reasoning and code generation tasks. To train the code PRM, we collect a function-level LeetCode dataset. We effectively integrate the PRM into the standard LLM inference process, achieving improvements over greedy search without additional inference overhead by token-level guidance. Our experiments on widely used datasets demonstrate robust performance with reduced computational costs. Furthermore, we analyze model transferability and generalization, showing that ASPRM exhibits both rating position, in-domain, and cross-domain generalization. We also find that combining data from different domains further enhances PRM performance. Lastly, our feature analysis of the AdaptiveStep division confirms its effectiveness and informativeness.

## Acknowledgement

We sincerely thank Zilin Zhu for providing valuable suggestions on efficiency optimizations of our code and Di Yang, Xiaochen Zhu, and the reviewers for their advice during the completion and review of the work. This work is sponsored by the Shanghai Science and Technology Commission Blockchain Special Project (No. 24BC3200100).

## Impact Statement

AdaptiveStep is an automatic, highly informative, and effective method for dividing reasoning steps. It can be easily applied to a wide range of complex tasks across various domains, such as code generation (as demonstrated in our paper) and AI-driven scientific reasoning. Furthermore, our exploration of the properties of AdaptiveStep PRM and its training data features will contribute to advancing process reward assignment in LLMs, potentially shaping the development of more general PRMs.

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

# A. Appendix

### A.1. feature statistic

In this part, we present the statistics of the decision token types in our dataset. Table 7 and Table 8 shows the statistical information of the math data, and Table 9 shows that of the code. We adopt en_core_web_sm model from Spacy library as tokenizer and POS tagger to make statistics. We show the cases for types of tokens in Appendix A.3.

### A.2. Dataset Information Statistic

In Figure 7, we report the statistical information of math training data for our dataset and ER-PRM, Math-Shepherd, PRM800K[2]. In Figure 8, we show the statistical information for our math BoN candidates.

### A.3. Case Study

We show divided cases in Math (Table 11) and Code domain (Table 12).

### A.4. Traning Data Distribution

Since the training data is generated and divided by the model, we will worry about whether there are a large number of difficult questions divided into few steps. Therefore, we analyze the relationship between step division and correct answer rate in our Mistral-generated GSM8k training data, we show the results in Figure 9. We find that only a few difficult questions (1.62%) is hard to answer by the training data generator with a few divisions.

Table 7: MetaMath-Mistral generated data statistic results: percentage of tokens types and percentage of decision tokens types for math domain. **Natural Sentence** stands for a piece of text separated by a *New line break* or *Punctuation* like Period and Question Mark. **Reasoning** represents *symbolic reasoning* or *Math Formula*; **Entity** represents *Noun* like apple or personal name; **Semantics** represents *Conjunction*, *Verb* and *Determiner*. We also find that there are few word level splits represented by **Split Word**; we retained these segmentation points to enhance the model's generalization at these points during PRM training.

| Categories | Subtypes | Position | |
| --- | --- | --- | --- |
| | | Token type proportion (78m) | Decision token proportion (1517k) |
| Natural Sentence | New line break | 3.85% | 2.70% |
| | Punctuation | 26.92% | 4.61% |
| Reasoning | Symbolic Reasoning | 15.39% | 6.79% |
| | Math Formula | 3.85% | 21.03% |
| Entity | Noun | 15.38% | 11.01% |
| Semantics | Conjunction | 20.51% | 29.00% |
| | Verb | 6.41% | 5.34% |
| | Determiner | 7.69% | 2.64% |

---

[2]Same to (Wang et al., 2024a), We counted the number of samples for PRM800K and is a quarter of that of Math-Shepherd.

Table 8: Proportion of decision tokens in the original data of the same type for math domain generated by MetaMath-Llama.

| Categories | Subtypes | Position | |
| --- | --- | --- | --- |
| | | Token type proportion (81m) | Decision token proportion (1413k) |
| Natural Sentence | New line break | 2.47% | 6.69% |
| | Punctuation | 28.40% | 14.91% |
| Reasoning | Symbolic Reasoning | 16.05% | 5.66% |
| | Math Formula | 3.7% | 20.24% |
| Entity | Noun | 14.82% | 7.35% |
| | Conjunction | 20.99% | 23.48% |
| Semantics | Verb | 6.17% | 5.24% |
| | Determiner | 7.4% | 2.99% |

Table 9: Proportion of decision tokens in the original data of the same type for code domain

| Categories | Subtypes | Position | |
| --- | --- | --- | --- |
| | | Token type proportion (17m) | Decision token proportion (47k) |
| Syntax Symbol | New line break | 6.99% | 11.79% |
| | Space Character | 77.58% | 1.60% |
| Numbers | Number | 4.21% | 0.84% |
| Logical Operators | Boolean Operators | 0.26% | 3.21% |
| | Arithmetic Operators | 2.04% | 3.54% |
| Definition | Def / Class | 0.53% | 1.82% |
| Import Statement | From / Import | 0.58% | 0.76% |
| Function | Type Defination | 0.16% | 0.48% |
| | Build-in Function | 0.49% | 0.77% |
| | Instance Method | 0.09% | 0.26% |
| Control Statements | If / Else / Elif | 0.64% | 3.51% |
| Loop Statements | For / While | 0.62% | 1.73% |
| Others | Return | 0.68% | 0.58% |
| | Punctuation Mark | 4.99% | 6.52% |

Table 10: Proportion of decision tokens in Code and Code Comment

| Categories | Trigger type(234k) | Token type (17m) | Line number(1599k) |
|---|---|---|---|
| Code | 47k (19.95%) | 4m (26.15%) | 1280k(80.02%) |
| Code comment | 187k (80.05%) | 13m (73.85%) | 319k(19.98%) |

Table 11: Samples of decision tokens for math domain.

| Categories | Subtypes | Sample |
|---|---|---|
| Sentence | New line break | works on 4 of them each day.\nAfter 5 days, |
| | Punctuation | If Billie has 18 crayons, and Bobbie has three times |
| Reasoning | Text reasoning | gives them 3 points. So in total, Joe's team has 3 + 3 = 6 |
| | Math formula | so x + 4x - 10 = 25 |
| Entity | Noun | Ron gets to pick a new book 1 out of 13 |
| Semantics | Conjunction | their ages is 34, so we can write the equation L + (L + 4) = 34. |
| | Verb | In 14 days, each dog will eat 250 grams/day |
| | Determiner | we can round this to the nearest whole number. |

Table 12: Samples of decision tokens for code domain

| Categories | Subtypes | Sample |
|---|---|---|
| Syntax Symbol | New line break | \n i += num_bytes |
| | Space Character | dp[i][j] += dp[i - 1][j] * (j - k) \s |
| Numbers | Number | j = (target - x * 2) // 2 |
| Logical Operators | Boolean Operators | if c in count and c != a: |
| | Arithmetic Operators | dp = [[0] * (n+1) for _ in range(n+1)] |
| Definition | Def | def is_valid(r, c): |
| | Class | class Solution: |
| Import Statement | From | from collections import defaultdict |
| | Import | import collections |
| Function | Type Defination | for size in list(dp[curr_sum]): |
| | Build-in Function | if abs(next_count + 1) ¿ 0: |
| | Instance Method | self.count = 0 |
| Control Statements | If | if len(tokens) ¡ 4: |
| | Else | else: |
| | Elif | elif level == 0 and expression[i] == ' ': |
| Loop Statements | For | for i in range(len(fronts)): |
| | While | while x != self.parent[x]: |
| Others | Return | return (merged[n // 2 - 1] + merged[n // 2]) / 2.0 |
| | Punctuation Mark | digit_sum = (l1.val if l1 else 0) + (l2.val if l2 else 0) |

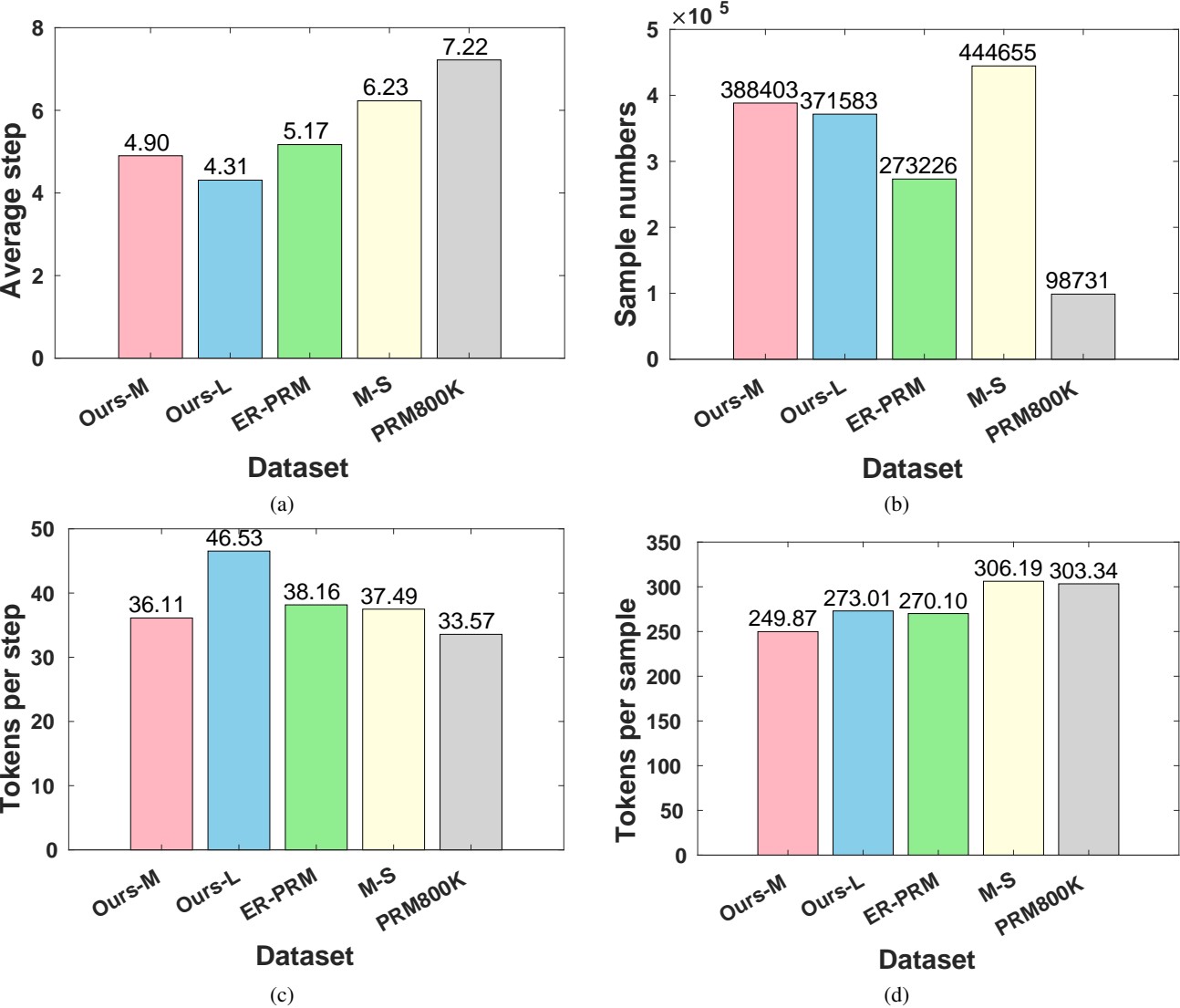

Figure 7: Statistic Information of our math dataset, Ours-M represents data constructed by Mistral, and Ours-L represents data constructed by Llama. ER-PRM, Math-Shepherd (M-S), PRM800K. (a): Average step; (b): Sample number; (c): Average tokens per reasoning step; (d): Sample length. We use a Mistral tokenizer for statistics.

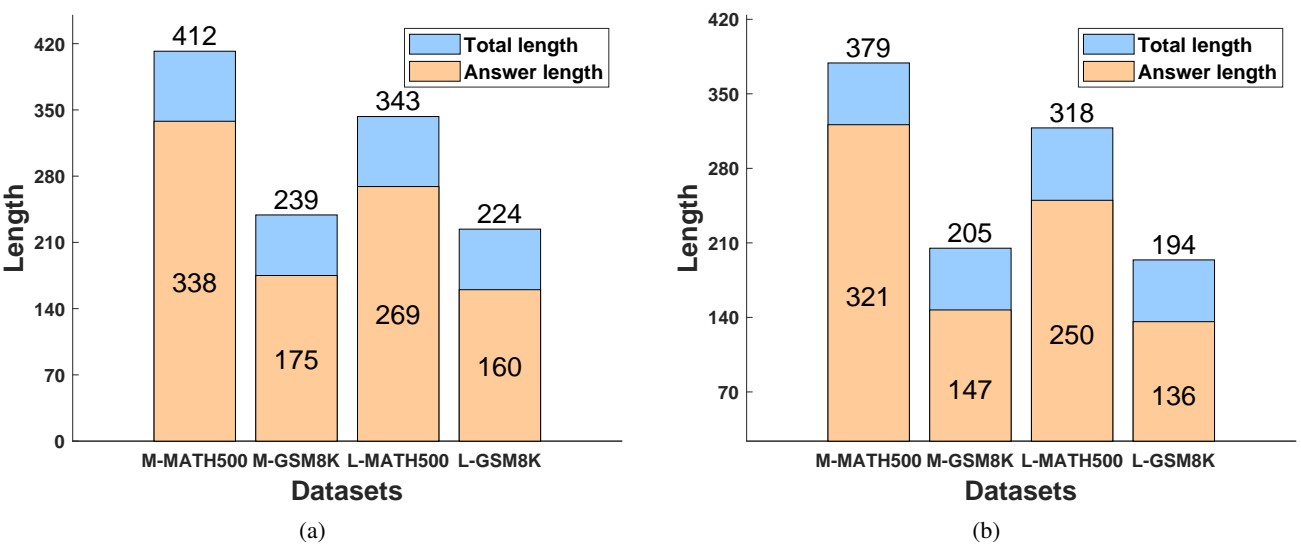

Figure 8: Statistic Information of our BoN dataset (a): Statistic with Mistral tokenizer; (b): Statistic with Llama tokenizer.

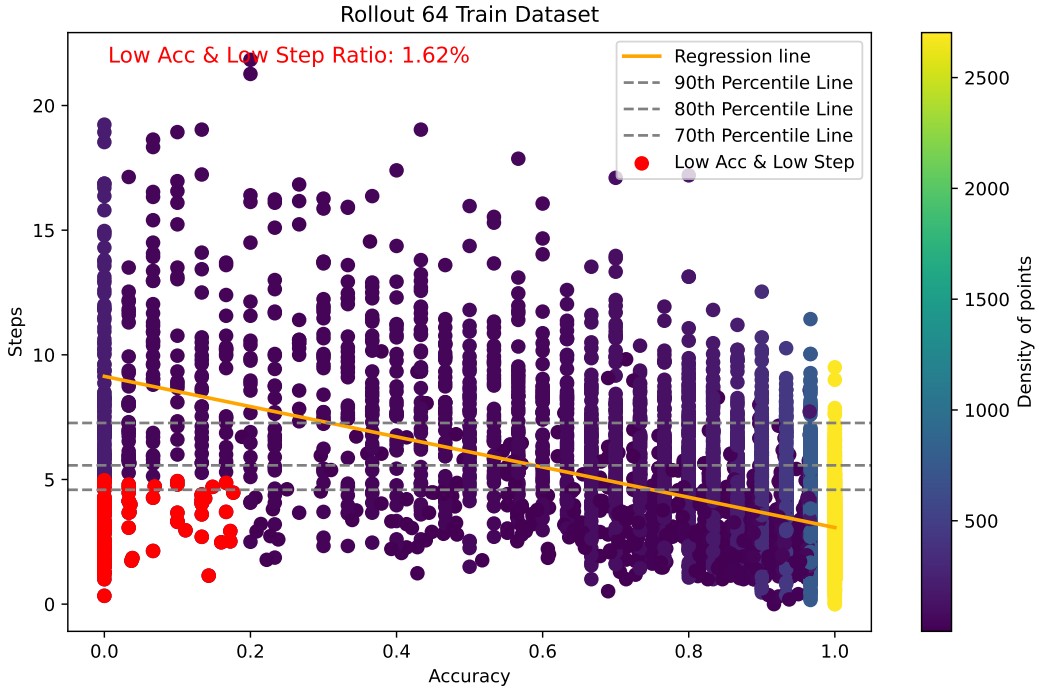

Figure 9: Training data distribution (division numbers and rollout accuracy).

