# OpenReview forum: "AdaptiveStep: Automatically Dividing Reasoning Step through Model Confidence"
_ICML.cc/2025/Conference — ICML 2025 poster_

### Official Review · Reviewer_NpJW · 2025-03-06

**Overall Recommendation:** 4

**Summary:**

This work proposes a method (Adaptive Step Process Reward Model - ASPRM) to split reasoning chains into reasoning steps based on the model confidence rather than pre-defined rules. For each token in the generated output, if the model’s probability for that token is below some threshold, then this token is treated as the start of a new reasoning step.

The authors train their ASPRM with rollouts and hard estimation of intermediate step values, and show that this adaptive step strategy yields higher-performing PRMs on math and coding benchmarks.

This work also experiments with token-level value guided decoding: during decoding, when the model generates a low-confidence token, the PRM can be used to rank the top M low-confidence tokens at that step. Experiments show that value-guided decoding performs best with ASPRM.

Eventually, multiple ablation studies are made in the experimental section to show the generalization capacity of the proposed approach.

## update after rebuttal

**Claims And Evidence:**

Yes, claims are supported by convincing evidence.

**Essential References Not Discussed:**

No critical related work missing to the best of my knowledge.

**Experimental Designs Or Analyses:**

Experiments and their analysis seems reasonable and valid. A lot of experiments have been made which is nice. It can get a little complicated to follow sometimes because in PRMs there are 3 models interacting with one another:
1. the model that generates trajectories and that is used with rollouts to generate data of the form `(partial trajectory, target score)`.
2. the actual PRM that is trained on the data generated in (1).
3. a policy model that is generating trajectories with the help of the PRM at inference time.

For each of these models the authors experimented with Mistral & Llama for math benchmarks, and deepspeed for LeetCode problems.

**Methods And Evaluation Criteria:**

Yes, the proposed methods and benchmarks (Math500 & GSM8k & LeetCode problems) make sense for introducing a new type of PRM

**Other Comments Or Suggestions:**

Clarification suggestions:

- In tables 2, 4, 5, 6 it would be clearer to put the performance of the method you compare against directly in the caption (or as an additional table row) so that the reader doesn’t have to search where this number is in another result table.
- The up & down arrows are nice (though the colors could be inverted: red for decline & green for improvement).

Typo:
- in the last paragraph of section 4.4, “_ results in 4.3 and 4.3 indicate …_”

**Other Strengths And Weaknesses:**

# Strengths

This paper presents a novel strategy to split reasoning chains into reasoning steps in the goal of training Process Reward Models. Throught numerous experiments the proposed method is shown to perform better than more traditional PRMs.
In addition, the method is tested as a value-guided decoding tool, which is also shown to perform well.
Eventually, numerous generalisation experiments and analysis is being presented at the end of the paper, showing rigorous evaluation.

# Weaknesses

One weakness of the proposed approach is that we need to define a confidence threshold to define “low” confidence steps and thus break the reasoning chain into steps. The decision of this threshold could influence the performance of the proposed method. The authors set the threshold such that 2% of reasoning tokens would fit under. This feels arbitrary and could benefit from further motivation.

In addition, an ablation study on the decision of the threshold could also benefit this paper. How does 2% compares to 1%, 5%, 10%, 20%, etc…

Eventually, the model used to estimate the confidence could also influence this decision: maybe some models are more confident than others and their threshold should be higher. An informed discussion with experiments could boost the quality of this work.

**Questions For Authors:**

No additional questions for now.

**Relation To Broader Scientific Literature:**

Process Reward Models (PRM) became a popular research topic recently because of their advantages over Outcome Reward Models (ORM) in giving intermediate feedback while training LLMs. This particularly became important as modern LLMs output reasoning chains before their final answer. Cheking the validity of each reasoning step can boost their performance. The idea of splitting reasoning chains based on model confidence (rather than on every new line character which is the default strategy) is novel to the best of my knowledge and seems to perform well. This shows once again that letting the model decide is better than imposing human judgement.

**Theoretical Claims:**

No theoretical claims found.

---

> ### Author Rebuttal · Authors · 2025-04-01
>
> # Response to Reviewer Comments
>
> **Dear Reviewer NpJW:**
>
> We would like to express our sincere gratitude for your time, your thorough review and valuable feedback on our manuscript. Your comments have provided us with important insights that will help improve the quality of our work.
>
> >## W1: Ablation Study on Threshold Decision
>
> **R1:** We appreciate your suggestion regarding the threshold decision process. Though our approach is grounded in cognitive theory, an ablation study is needed in the revised manuscript to demonstrate the impact of different thresholds. We add the BoN results of ASPRM models trained with threshold of 0.5%, 1%, and 1.5%. However, larger thresholds (3%, 5%, 10%) mean perform more rollouts. Due to computational resource constraints during rebuttal, we are unable to conduct extensive experiments with various models and higher threshold combinations, we will conduct further ablation studies when we have more computational resources available. Table 1 and Table 2 shows the BoN results of ASPRM trained with threshold at 0.5%, 1.0%, and 1.5%, compared with the baselines, and a 0.5% change means that each solution has one less step on average. We find that although there is some fluctuation, a greater number of segments within the range of 0.5% to 2% generally means better judging capability. We used more models (MetaMATH-Llama-7b / -13b - 70b) for generation to supplement the results. We hope for your understanding and approval due to the lack of additional results.
>
> **Table 1: Bo64 results on Math500.**
>
> | model | MetaMATH-Llama-7b | -13b | -70b |
> |:---:|:---:|:---:|:---:|
> | ASPRM-M-T0.5 | 25.00 | 28.80 | 32.60 |
> | ASPRM-M-T1.0 | 25.00 | 29.60 | 31.60 |
> | ASPRM-M-T1.5 | 27.80 | 28.80 | 31.40 |
> | ASPRM-M-T2.0 | 25.40 | 31.00 | 34.60 |
> | Math-shepherd | 28.40 | 31.00 | 33.00 |
> | ASPRM-L-T0.5 | 31.80 | 35.2 | 37.00 |
> | ASPRM-L-T1.0 | 31.60 | 34.80 | 36.20 |
> | ASPRM-L-T1.5 | 32.00 | 35.60 | 38.60 |
> | ASPRM-L-T2.0 | 33.40 | 37.80 | 40.00 |
> | ER-PRM | 33.20 | 37.40 | 38.80 |
>
> **Table 2: Bo64 results on GSM8k.**
>
> | model | MetaMATH-Llama-7b | -13b | -70b |
> |:---:|:---:|:---:|:---:|
> | ASPRM-M-T0.5 | 79.45 | 83.32 | 85.82 |
> | ASPRM-M-T1.0 | 81.04 | 83.55 | 88.48 |
> | ASPRM-M-T1.5 | 81.65 | 82.64 | 89.61 |
> | ASPRM-M-T2.0 | 81.27 | 85.80 | 89.23 |
> | Math-shepherd | 84.23 | 85.22 | 88.40 |
> | ASPRM-L-T0.5 | 86.84 | 86.85 | 91.38 |
> | ASPRM-L-T1.0 | 89.60 | 86.58 | 91.74 |
> | ASPRM-L-T1.5 | 86.20 | 90.00 | 90.02 |
> | ASPRM-L-T2.0 | 85.52 | 88.25 | 91.66 |
> | ER-PRM | 86.58 | 87.49 | 88.86 |
>
> Besides, we also investigated the influence of various thresholds on the performance of TVD. The results have been collated and presented in the following two figures: [Figure 1](https://anonymous.4open.science/r/PIC-0969/fig2.png) and [Figure 2](https://anonymous.4open.science/r/PIC-0969/fig3.png).
>
> >## W2 : Model Influence of Confidence Threshold
>
> **R2:** We use the pecentage of whole distribution of model confidence, instead of a fixed threshold, this means that different models will indeed have different threshold values, and different tasks will also have different threshold values. We use a table to illustrate this, the following Table 3 below shows the values at a 2% confidence threshold for different models and tasks. We find that for the same split ratio, models with greater capabilities and simpler tasks tend to have higher threshold values.
>
> **Table 3: 2% confidence threshold for different models and tasks.**
>
> | dataset | MATH500 | GSM8k |
> |:---:|:---:|:---:|
> | MetaMATH-7b | 25.51 | 34.26 |
> | 13b | 25.78 | 35.91 |
> | 70b | 33.41 | 39.51 |
>
>
>
> >## W3 and T1: Figure, Table and Typographical Errors
>
> **R3:** Thank you for highlighting the issues with the table, the up and down arrows' color and the typographical errors in our manuscript. We will correct these inaccuracies in the revised version to improve clarity and readability.
>
> ---
>
> **Thank you again for your time, thorough review and constructive feedback. Your insights have significantly helped us identify areas for improvement in our work. We welcome any additional suggestions that could further enhance the quality and rigor of our research.**

---

### Official Review · Reviewer_LC8A · 2025-03-13

**Overall Recommendation:** 3

**Summary:**

The paper addresses the challenge of training Process Reward Models (PRMs) for large language model reasoning by introducing a novel step segmentation method called AdaptiveStep. Instead of using fixed rules or token counts to break a model’s chain-of-thought into steps, AdaptiveStep dynamically segments the reasoning process based on the model’s own confidence in predicting the next token.

The authors sample solution paths from an LLM and compute the probability (confidence) for each generated token; tokens with unusually low confidence (below a learned threshold τ) are treated as decision boundaries, initiating a new reasoning step​. Using these segmented steps, the authors then train a PRM by simulating rollouts from each partial solution (each step) to see if a correct final answer can still be reached​.
Each step is labeled positive if any continuation yields a correct answer, or negative if all continuations fail, following a heuristic “hard” reward assignment, similar to how PRM is usually trained.

AdaptiveStep-augmented PRMs (termed ASPRM) are tested on complex reasoning tasks in mathematical problem solving (GSM8K, MATH dataset) and code generation (LeetCode-style programming problems, LiveCodeBench) and the approach achieves good results overall.

## update after rebuttal

I have given a weak accept to the paper and I wont mind seeing it get accepted.

**Claims And Evidence:**

A lot of claims are made in the paper and are backed by empirical results.

1. Dynamically segmenting reasoning steps by model confidence yields more “decision-making insights” per step than naive rule-based segmentation: This is supported qualitatively by examples and quantitatively by analysis of the segmentation output: they observe that AdaptiveStep tends to insert breaks at meaningful junctures (e.g. mid-formula or before a crucial choice in logic) rather than at arbitrary punctuation or fixed intervals​.

2. The claim that no manual annotation is needed is right as the proposed method relies purely on the model’s probabilities and an automated rollout procedure.

Other claims like lower cost and reduced number of samples are also verified in the paper.

**Essential References Not Discussed:**

I think the authors mentioned all PRM related past works. One area that wasn’t explicitly mentioned is the line of research on verifier models or consistency checks aside from PRMs. For example, works like Cobbe et al. (2021) or Li et al. (2022) on verifying chain-of-thought or Iterative Polishing might be related.

**Experimental Designs Or Analyses:**

The experiments are well designed and the authors clearly describe their experiment setup, including datasets, model choices, baselines, and metrics used to test the models. The baselines themselves are appropriate and the authors make an effort to ensure comparability. For math, they use published open-source PRMs (Math-Shepherd and ER-PRM). It’s a little unclear whether they recomputed those baselines’ performance on their setup or took numbers from the literature. Maybe I might have missed this in the paper.

The metrics and analysis of results are appropriate. They measure both Accuracy/Pass@1 (to see if the PRM hurts single-shot performance – it doesn’t; guided decoding often improves it) and Best-of-N accuracy (to demonstrate how well the PRM can identify a correct solution among many).

On thing missing in the paper is that the paper doesn’t report any significance tests. Improvements like +3% on GSM8K could be within error margin if not enough samples. I am not sure what % of improvement is good.

**Methods And Evaluation Criteria:**

The proposed method is very simple and using loprobs to estimate confidence is not new and has been explored a lot in the past. Applying this to PRM is unique but this approach makes intuitive sense – low confidence often indicates the model is choosing among multiple possibilities (e.g. figuring out the next step in a math proof or deciding on a coding approach), which is exactly where a new step and potentially a feedback signal would be most valuable.

The PRM training procedure – labeling each segmented step via rollouts – is a logical way to obtain supervision. It mirrors prior work (like Wang et al., 2024a’s heuristic rollout for Math-Shepherd) but importantly removes the need for manual identification of step boundaries, which is one of the main contributions. The training is the same as past works and the authors’ choice to use binary labels (hard estimation) for each step is reasonable and aligned with previous PRM approaches.

The evaluation criteria and benchmarks are appropriate and standard for the domain - math and code. In the code domain, since no established stepwise PRM existed publicly, they construct a baseline Outcome Reward Model (ORM) by training a reward model that only gives feedback at the final answer​.

Overall, I would say that the benchmarks and metrics clearly align with the paper’s goals and is pretty standard.

**Other Comments Or Suggestions:**

-

**Other Strengths And Weaknesses:**

### Strengths
- The core idea of using the model’s own confidence to segment reasoning steps is very clever. It addresses a clear limitation of prior methods, which relied on ad-hoc rules or costly annotations.
- AdaptiveStep yields a more efficient data generation process for training the PRM. By segmenting only when needed, it produces far fewer total steps to label.
- The experiments follow past works and results are provided on math and code. The authors provide insightful analysis of where the model places step breaks (e.g., highlighting that conjunctions and math operators often trigger low confidence​).

### Weakness:
- AdaptiveStep’s segmentation is only as good as the model’s confidence estimations. If an LLM has idiosyncrasies in its probability outputs (e.g., it might be overconfident in certain wrong steps or underconfident in trivial but rare phrasing), the segmentation could be suboptimal.
- A notable weakness is the somewhat arbitrary choice of using the bottom 2% confidence as the segmentation threshold. While the authors cite a cognitive science justification, this parameter was not deeply examined.
- [MINOR]: Interesting to see how the method would perform in domains without binary success criteria (e.g., open-ended logical reasoning or commonsense questions where “correctness” is fuzzy). The paper’s approach relies on having a ground truth check to label steps.
- The baselines chosen were appropriate (other PRMs and an outcome model), but one could argue that the paper doesn’t compare against the strongest possible alternatives. For example, Reinforcement Learning from Human Feedback (RLHF) at the step level or even outcome level could improve reasoning – how would a model fine-tuned with RLHF on these tasks compare to using a PRM? Similarly, OmegaPRM (which uses MCTS and presumably a powerful orchestrator) is mentioned but not empirically compared.
- [MINOR] A minor weakness is that some details of the method are not fully spelled out in the paper, potentially hindering replication. For example, the exact number of samples generated per question, the value of J (number of rollouts per step) used for labeling, or the training hyperparameters for the PRM model are not explicitly listed (at least not in the excerpt we see).

**Questions For Authors:**

1. Your method assumes the model’s confidence (top-1 probability) is a reliable indicator of decision difficulty. Did you observe any cases where the model was overconfident in a wrong step (thus no break inserted) or underconfident in an easy step (inserting an unnecessary break)? In other words, how robust is AdaptiveStep to calibration errors in the base LLM’s probabilities?
2. How sensitive are your results to the choice of the 2% confidence threshold for segmenting steps? Did you experiment with different percentages or adaptive thresholds per dataset/model? It would be useful to know if 2% is truly optimal or just one reasonable setting.
3. Could you clarify the data generation process for PRM training in terms of scale? Specifically, how many solutions N did you sample per problem to determine the confidence distribution and threshold, and how many rollouts J per step were used to label each step? These numbers are important to understand the compute cost.
4. For the math baselines (Math-Shepherd and ER-PRM), did you reimplement/retrain those methods on the same datasets and base models that you used for ASPRM, or are you citing their reported results?
5. You cite OmegaPRM (which uses MCTS) and stepwise RLHF approaches. How do you expect AdaptiveStep to compare to those in practice?
6. How would AdaptiveStep and PRM training apply to domains where checking correctness is non-trivial (for example, commonsense reasoning or legal question answering where there isn’t a single numeric answer or test cases)?

**Relation To Broader Scientific Literature:**

The authors discuss the usefulness of PRMs and discuss prior works. There is a good coverage of recent literature on PRMs and stepwise reasoning alignment.
One interesting thing is that the authors used OpenAI cookbook ideas to assess logprobs, which is interesting. I wonder if the authors are the first to turn this idea into a full pipeline for training a reward model.

**Theoretical Claims:**

The paper is mostly empirical and has no theoretical claims as such.

One interesting claim is the rationale behind the 2% threshold for low-confidence tokens. The authors justify setting the threshold so that roughly 2% of generated tokens are below it, citing Kahneman’s (2011) finding that about 2% of human thinking is “deep thinking”​. This is very interesting but hard to judge. The paper does not prove that 2% is the best choice; rather, it assumes this fraction yields a reasonable number of decision points. While the results with 2% are good, I would be interested in seeing a grid search over more possible values.

Finally, the main claim of the paper is that “low model confidence indicates a potential decision point”. Intuitively, this claim is sound – if an LLM is uncertain about the next token, it likely means multiple continuations are plausible, implying a branch in reasoning. The paper’s empirical analysis supports this (confidence dips align with meaningful junctures), but theoretically one could question if low confidence always corresponds to an important decision. There might be cases where an LLM’s confidence is low simply because it’s generating an uncommon word or name, not because it’s at a logical decision point.  Therefore, this claim is plausible and supported by examples, but not theoretically guaranteed for all scenarios.


One theoretical aspect that might have been explored more is the calibration of model confidence. AdaptiveStep assumes the model’s token probability is a meaningful indicator of uncertainty. In theory, if a model’s confidence estimates are poorly calibrated, the threshold might not accurately reflect decision difficulty. The paper doesn’t delve into calibration theory or prove that the chosen LLMs have well-calibrated confidences in these domains. They proceed empirically, and indeed the approach works, implying the confidences were informative enough.

---

> ### Author Rebuttal · Authors · 2025-04-01
>
> # Response to Reviewer Comments
>
> **Dear Reviewer LC8A:**
>
> We greatly appreciate your insightful comments and suggestions and thank you for your time. Due to the character limit, we have summarized your questions. If we have missed or misunderstood any points, please let us know.
>
> >## Related Work
>
> Thank you for your valuable suggestions regarding related work. We will incorporate literature on reward models for LLMs in our revised manuscript, including BT model, GenRM from Zhang et al. (2024), Pairwise RM from Jiang et al. (2023), Cobbe et al. (2021), Li et al. (2022) and so on.
>
>
> >## Q1 and W1: Suboptimal Concerns
>
> **R1:** Thank you for expressing your concerns. We provide a [statistical results](https://anonymous.4open.science/r/PIC-0969/fig1.png) to show the relation between segmentation numbers and task difficulty. As shown in the figure, there are indeed some segmentation issues in the lower-left corner and on the right side, where few difficult problems are segmented into fewer steps, or certain easy problems are segmented into more steps. Although these cases are rare (fewer than 2%), the issue does exist. We believe that combining rule-based methods with AdaptiveStep would result in better divisions.
>
> >## Q2 and W2: Threshold Settings
>
> **R2:** Thank you for your insightful suggestion. We will include more percentage settings in our experimental evaluation to provide a more comprehensive analysis of our method's performance across various conditions. We used more models for generation to supplement our results. [Table 1](https://anonymous.4open.science/r/PIC-0969/fig4.png) and [Table 2](https://anonymous.4open.science/r/PIC-0969/fig5.png) shows the BoN results of ASPRM with threshold at 0.5%, 1.0%, and 1.5% compared with the baselines, and a 0.5% change means that each solution has one less step on average. Due to computational limitations, we are unable to scale up more rollouts in a short time, but we will conduct further analysis in subsequent versions, we hope for your understand. We find that although there is some fluctuation, a greater number of segments within the range of 0.5% to 2% generally means better evaluation capability.
>
> >## Q3, Q4 and W6: Implementation Details
>
> **R3:** Thank you for highlighting the missing methodological details. Regarding data preprocessing, we have provided training data specifications in Sec.4 (Parameter Setting), which documents 30 solutions per problem and 8 rollouts per step for labeling. In Sec.4.4 (Construction Efficiency), we present a comparison of computational costs with others.
>
> For ASPRM training, we employed a batch size of 256 and learning rate of 1e-6 (Mistral), 2e-6 (Llama), 5e-6 (Deepseek), consistency with the Math-Shepherd parameters in the OpenRLHF script.
>
> For baselines, we utilized the released models of other methods and rigorously adhered to their prescribed usage for evaluation on our dataset. At the 7B model scale, our reported Math-Shepherd BoN results exceed those in the original paper, demonstrating the fairness of our comparisons.
>
> >## Q5 and W4: OmegaPRM and RLHF
>
> **R4:** Thank you for raising these questions. OmegaPRM is a method that integrates binary search into MCTS to construct data, while ours is a fine-grained step segmentation method. The methods are orthogonal, and since this and data have not been released, we did not use it as a baseline for comparison. RLHF is indeed a great comparison approach, but the computational resources required for RLHF are currently beyond our capacity. Therefore, we opted for the more lightweight BoN and TVD methods for evaluation, which are commonly used in previous reward model works.
>
> Regarding the combination with these methods, we believe that fine-grained segmentation helps OmegaPRM more accurately identify errors, thereby improving the error detection efficiency. As for RLHF, positions with low confidence indicate that the model likely generate other branches. Compared to rule-based step-level RLHF, we believe that using AdaptiveStep allows the model to better learn from these branches, leading to good results. We will also validate these claims when sufficient computational resources are available.
>
> >## Q6 and W3: Commensense Domain
>
> **R5:** Utilize to commensense domains remains challenging for process reward models. Some researchers employ powerful models like GPT-4o or human judgment for evaluation [1]. We believe AdaptiveStep offers advantages in this context: our segmentation method identifies low-confidence model outputs, suggesting more choices and higher error probability, thereby improving labeling efficiency.
>
> **[1]** O1 Replication Journey: A Strategic Progress Report: Part I
>
> ---
>
> **We sincerely thank you for your constructive feedback and insightful recommendations. These comments have significantly helped us identify areas for improvement in our manuscript. We look forward to incorporating more suggestions from you to enhance the quality and clarity of our research.**

---

### Official Review · Reviewer_ZPdL · 2025-03-14

**Overall Recommendation:** 3

**Summary:**

1. In this paper, a novel step dividing method, AdaptiveStep is proposed. The method enables automatic step dividing while being more informative than rule-based step dividing method.
2. By adapting AdaptiveStep, the ASPRM demonstrates stronger discriminative power at the token level than existed method at some extent.
3. The author open-sourced a LeetCode dataset, which may benefit following works on PRM.

**Claims And Evidence:**

The article claims that ASPRM is a SOTA PRM model in the introduction, but in the experiment, the ASPRM-M model is not as effective as Shepherd-M. In addition, the article does not show the effect of ASPRM in reinforcement learning training. In view of these circumstances, I think the conclusion that ASPRM is SOTA is problematic.

**Essential References Not Discussed:**

I believe essential references are discussed. However, it would be better to discuss in the related works part how this work differ from the previous works to help the reader gain a big picture of this work.

**Experimental Designs Or Analyses:**

The paper lacks experiments and results on how PRM can improve the final performance of the model when used for reinforcement learning. In terms of experimental design, it is inappropriate to compare Llama based PRM with Mistral based PRM.

**Methods And Evaluation Criteria:**

I think the proposed method make sense to me. Using confidence score as an indicator to identify difficult tasks and divide step based on it seems like a feasible solution to balance cost and informative of the annotated dataset.

**Other Comments Or Suggestions:**

Regarding the presentation of TVD results across different methods, I suggest replacing the current large table containing many blank cells (where code models are not applicable to certain tasks) with a series of focused bar charts.

**Other Strengths And Weaknesses:**

The weaknesses mainly on the experiment part. Results on more scenarios (performance lifting when leverage the proposed PRM on RL training), more based models (32B, 70B models) are needed to fully exhibit the effectiveness of the proposed method.

**Questions For Authors:**

1. I noted that the RPM training and TVD methods described in this paper have similarities to previous work in the field. Could the authors clarify the specific novel contributions or advancements in these two areas? In particular, I would appreciate a more detailed explanation of how these approaches differ from or improve upon existing techniques, as this would help better position the work within the current literature and highlight its unique contributions.
2. I appreciate the contribution of open-sourcing the LeetCode dataset, but I have some concerns about the ethical implications. Given that LeetCode problems are proprietary content, could the authors clarify the licensing arrangements or permissions obtained for redistributing this data? Additionally, it would be helpful to understand what steps were taken to ensure compliance with relevant terms of service and intellectual property rights when creating and sharing this dataset."

**Relation To Broader Scientific Literature:**

### Innovation in Reasoning Step Segmentation
Existing Process Reward Models (PRMs) typically use rule-based methods, such as predefined symbols or fixed-length reasoning steps, to segment reasoning steps. The AdaptiveStep method proposed in this paper automatically segments reasoning steps based on the model's confidence in predicting the next token. This aligns with the cognitive cost theory, which suggests that the cognitive cost of reasoning depends on task difficulty. Additionally, many reasoning errors stem from incorrect numerical calculations or misuse of words, further supporting the necessity of segmenting reasoning steps at critical points.

### Improvement in Process Reward Models (PRMs)
The AdaptiveStep PRM (ASPRM) introduced in this paper demonstrates excellent performance in mathematical reasoning and code generation tasks, outperforming existing open-source PRMs. This is consistent with research emphasizing the importance of intermediate reasoning steps in complex tasks and demonstrating how step-by-step feedback enhances reasoning reliability and reduces logical errors.

**Theoretical Claims:**

This is primarily an experimental paper that focuses on empirical results rather than theoretical claims requiring formal proofs. Therefore, no formal proofs needed to be verified during my review.

---

> ### Author Rebuttal · Authors · 2025-04-01
>
> # Response to Reviewer Comments
>
> **Dear reviewer ZPdL:**
>
> Thank you for your insightful comments and valuable feedback. Below, we address the concerns raised and provide clarifications and improvements.
>
> >## W1: Concerns about experiments results
>
> **R1:**  Thanks for your comment and we will revise our paper to clarify the setting/scope of the SOTA statement accordingly. We argue that the overall performance of ASPRM-M exceed that of the baselines. Specifically, ASPRM-M significantly outperforms baselines across all TVD experiments. In the BoN-Mistral generated data, ASPRM-M performs better than Math-Shepherd when N is small. In positions with few instances, such as Figure 4(b) for Bo64 and Figure 4(d) for Bo16 and above, ASPRM-M is slightly worse than Math-Shepherd. In subsequent experiments, such as those involving position generalization, ASPRM also performs well.
>
> >## W2: Llama compared with Mistral
>
> **R2:**  Thank you for bringing this up. The experiments involving Llama and Mistral were simply presented in the same figure for space efficiency. There was no intention to directly compare the performance of these base models. We likely would not have conducted the experiments with the Mistral-based ASPRM-M model if we want to make a direct comparison.
>
> >## W3: Reward Model Related Work
>
> **R3:**  We appreciate the suggestion to add related work on reward models. We will revise the manuscript to include relevant references and discussions on prior work in this area (e.g., the Bradley-Terry model, GenRM, Pairwise reward model, etc.).
>
>
> >## W4: RLHF and More Models
>
> **R4:**
> **RLHF Methods:**
> While RLHF methods are often used to evaluate reward models, they are not strictly necessary for this area. Several influential works on (process) reward models have not included RLHF experiments but use BoN or test-time scaling to evaluate the model’s capability, yet they remain impactful in the field [1, 2]. Our focus was on proposing and testing a novel reasoning step segmentation method. Given the substantial computational resources required for RLHF experiments, we are unable to include them in this study. We kindly hope that the reviewer takes this into consideration.
>
> **More Models:**
> Regarding the inclusion of more model scales, we have conducted experiments on a wider range of model sizes and thresholds in [Table 1](https://anonymous.4open.science/r/PIC-0969/fig4.png) and [Table 2](https://anonymous.4open.science/r/PIC-0969/fig5.png), which we will incorporate into the updated manuscript, we observe that as the model size increases, the PRM’s ability to make judgments remains effective.
>
>
> [1] Let's Verify Step by Step
> [2] Training Verifiers to Solve Math Word Problems
>
> ---
>
> >## Q1: Our Contributions
>
> We sincerely request that the reviewer reconsider the contributions of our work. Our primary contribution is the introduction of a novel reasoning step segmentation method AdaptiveStep, which we validate in the PRM senario. The PRM trained with AdaptiveStep can guide the task model in generation and reasoning at the token level, which previous PRMs have not been able to do in our experiments. Moreover, we have successfully extended PRM to domains where step segmentation is difficult using rules (such as code generation). And we use a single model for data construction and reduces construction cost by 30% in the math domain and 60% in the code generation domain compared to rule-based methods.
>
> In the experimental setup, we have made every effort to maintain consistency with previous work, including model selection, PRM training methods, and evaluation techniques. This was done to establish a baseline for a fair comparison, which consequently limited the potential for introducing significant innovations in the training of the PRM or the testing of BoN/TVD.
>
> ---
>
> >## Q2: Credit and License
>
> Thank you for pointing this out. We use this dataset only for research purposes and will strictly regulate its usage context and license. We will do our best to align our open-source work with [LiveCodeBench](https://github.com/LiveCodeBench/LiveCodeBench), which collects data from LeetCode weekly contests.
>
> ---
>
> >## General Rebuttal
>
> We propose a novel reasoning step segmentation method and have evaluated it within the context of process reward models, supported by statistical analysis and extensive experiments. While RLHF methods are a reasonable evaluation approach, the heavy computational resources required for such experiments made it unfeasible for us to incorporate them, so we use the lightweight BoN and TVD method. As highlighted earlier, many impactful reward model studies have not employed RLHF experiments, yet they have been influential in the field.
>
> ---
>
> **We greatly appreciate the your thoughtful feedback and your time for upgrading our work once again, and we hope that these clarifications will be satisfactory. We welcome any further constructive comments that can help improve the quality of our work.**

---

### Official Review · Reviewer_T2AZ · 2025-03-15

**Overall Recommendation:** 3

**Summary:**

Simple yet effective method for segmenting reasoning traces into individual steps, resulting in modest but significant improvements on relevant math and coding benchmarks.
The paper proposes to segment reasoning traces according to next-token confidence levels, instead of e.g. heuristics like new lines.
The experiments show that this segmentation method outperforms prior work in PRM training and ultimately results in higher performance on Math and Coding datasets.
The analysis is limited to coding and math problems, which however constitute highly relevant domains.

**Claims And Evidence:**

Claim: AdaptiveStep yields a SOTA PRM for math tasks.
Evidence: Evaluation on relevant benchmarks and for multiple models.

**Essential References Not Discussed:**

NA.

**Experimental Designs Or Analyses:**

yes.

**Methods And Evaluation Criteria:**

The method constitutes cutting generations at tokens that have low probability, training a PRM on the data, and using the PRM to train a Math/coding model. this is a straightforward procedure that follows relevant prior work.

**Other Comments Or Suggestions:**

See weaknesses.

**Other Strengths And Weaknesses:**

Strengths:
Simple method and significant improvements on relevant benchmarks.
Paper is well written and problem setting is relevant.
Cross-domain generlaisation results suggest robustness.

Weaknesses:
- Failure cases of proposed segmentation method are neither discussed nor empirically analysed. It would be nice if the authors could both provide examples of common failure cases, as well as a discussion of them
- The Adaptive step method is only described in words, it would be great to have an algorithm definition in the methods section.

**Questions For Authors:**

See weaknesses.

**Relation To Broader Scientific Literature:**

The paper fits well with recent literature on step-by-step reasoning for math and coding problems.

**Theoretical Claims:**

NA.

---

> ### Author Rebuttal · Authors · 2025-04-01
>
> # Response to Reviewer Comments
>
> **Dear reviewer T2AZ:**
>
> We sincerely appreciate your constructive feedback and the time you have devoted to reviewing our manuscript. We try to address your concerns point by point.
>
> >## W1: Discussion or empirical analysis for failure cases of AdaptiveStep
>
> **Response:**  Thank you for highlighting this important aspect. Though explicitly classifying "failure cases" in the model's segmentation is challenging as what is regarded as "failure" for humans indeed aligns with the model's confusion points, we still identified several instances of suboptimal segmentation results.
>
> We understand that failure cases can be categorized from two perspectives. The first perspective involves words that should remain intact but are erroneously split into separate parts. The second perspective concerns instances where the model fails to segment at positions prone to errors. We illustrate the first type of failure cases through the following examples, while the second type is demonstrated through a figure.
>
> **Example 1:** "We can rewrite the **quad** / **ratic** as $y = 3(x^2 + 2x) + 9$"
>
> **Example 2:** "**Sub** / **stituting** these values" — where "quadratic" and "substituting" are incorrectly segmented despite being complete words
>
> **Example 3:** "**Natal** / **ie** would need to trade" — where "Natalie," a proper noun referenced in the question, is inappropriately segmented
>
> We conduct a statistical analysis of the aforementioned type where words are split, and find that about 2% of the segmentation belongs to this category. However, since this split is determined by the model itself, we retained these splits during the training process.
>
> Additionally, in the very begining of our work, we observed that approximately 3% of split points occur at the beginning of solutions, which we classify as erroneous segmentations, we removed these split positions because they indicate that the solutions have not yet started to be generated.
>
> For the second perspective, due to limitations of the base model, certain complex questions exhibit little segmentation in their solutions, as illustrated in the [figure](https://anonymous.4open.science/r/PIC-0969/fig1.png), we believe that the problems in the lower-left corner (the 1.62% with low accuracy after 64 generations and segmentations numbers lower than 5) represent problems that exceed the model's capabilities, and require additional supervisory information to assist.
>
> In summary, aside from the examples that are beyond the model's capabilities to judge (which account for about 5% of the total questions), there are 5% of the splits that, from a human perspective, seem unreasonable. We have removed some of these (those at the beginning of the solutions) and retained others. We will explain this in more detail in the appendix. Thank you again for your highlighting.
>
> >## W2: Algorithm definition
>
> **Response:** Thank you for your suggestion. We will include the following algorithmic in out next version:
>
> ## Algorithm: PRM Training from Confidence-Guided Rollouts
>
> **Inputs**:
> - `Q`: A sequence of questions
> - `N`: Number of responses to generate per question
> - `J`: Number of rollouts per split point
>
> **Output**:
> - A trained process reward model (PRM)
>
> ```text
> for each question q in Q do
>     // Step 1: Generate N responses for the question
>     responses ← GENERATE_RESPONSES(q, N)
>
>     // Step 2: Compute confidence distribution for the responses by eqution (1)
>     confidence_distribution ← COMPUTE_CONFIDENCE(responses)
>
>     // Step 3: Determine threshold and find split points
>     threshold ← COMPUTE_THRESHOLD(confidence_distribution)
>     split_points ← FIND_SPLIT_POINTS(confidence_distribution, threshold)
>
>     for each split in split_points do
>         labels ← empty list
>
>         // Step 4: Perform J rollouts for each split point
>         for i from 1 to J do
>             label ← PERFORM_ROLLOUT(split)
>             APPEND(labels, label)
>         end for
>
>         // Step 5: Aggregate rollout labels into a hard estimate by eqution (2)
>         prm_label ← HARD_ESTIMATE(labels)
>
>         // Step 6: Train the PRM with the obtained labels by eqution (3)
>         TRAIN_PRM(prm_label)
>     end for
> end for
> ```
>
> ---
>
> **We welcome any additional feedback you may have regarding our manuscript or this response. We are committed to incorporating your suggestions to enhance the quality of our work. Thank you again for your valuable insights.**

---

> > ### Comment · Reviewer_T2AZ · 2025-04-04
> >
> > The authors have addressed my concerns. I maintain my score as I believe that the contribution is significant, but do not believe that it merits a "strong accept" rating.

---

> > > ### Author Response · Authors · 2025-04-06
> > >
> > > Thank you very much for your time and for recognizing our contribution as “significant.” We truly appreciate your thoughtful feedback. We just want to clarify a small point in case there was any misunderstanding regarding the ICML scoring system: under the new criteria, a score of 3 corresponds to a “weak accept,” 4 to an “accept,” and 5 to a “strong accept.” Since you kindly mentioned that the work may not merit a “strong accept,” we were wondering if you might consider updating the score to a 4 (“accept”). Thank you again for your valuable comments.

---

### Decision · Program_Chairs · 2025-05-01

**Decision:**

Accept (poster)

**Comment:**

This paper propose a simple method for training a process reward model (PRM), deemed AdaptiveStep. In the proposed method, the intermediate steps are determined based on the confidence of the model using a threshold that has been chosen to force 2% of the reasoning steps to become break points. This is in contrast to prior work that uses heuristics such as line breaks as their break points. The paper shows that the proposed PRM achieves better performance compared to existing methods. The experimental section of the paper as well as the interpretation of the results seems solid and appreciated by the reviewers as well. During rebuttal, reviewers asked for further ablations and evidences that were successfully provided by the authors. There are some concerns about the claims where the proposed method does not always lead to better performance compared to other methods. I think the authors have done a good job trying to understand and explain these issues. I think it would be better if the authors also tone down the claims in abstract and introduction a bit to reflect the results. There is also concerns about lack of empirical evidence in using the new PRMs to train RL; I agree with authors that using the new models with BON to see the scaling is sufficient, especially for this type of rewards and given that BON is already a very strong baseline for RLHF problem (Beirami et al 2024). Overall, I think this is a solid paper that should be accepted to the conference. Congratulations to the authors!

Beirami, Ahmad, et al. "Theoretical guarantees on the best-of-n alignment policy." arXiv preprint arXiv:2401.01879 (2024).